# A Systematic Review for Effective Preventive Public Education of Respiratory Infection

**DOI:** 10.3390/ijerph18083927

**Published:** 2021-04-08

**Authors:** Woojae Han, Jeong-Soon Yu, Sihun Park, Myung-Soon Kwon

**Affiliations:** 1Laboratory of Hearing and Technology, Research Institute of Audiology and Speech Pathology, College of Natural Sciences, Hallym University, Chuncheon 24252, Korea; woojaehan@hallym.ac.kr (W.H.); psh940310@nate.com (S.P.); 2Division of Speech Pathology and Audiology, College of Natural Sciences, Hallym University, Chuncheon 24252, Korea; 3Research Institute of Nursing Science, School of Nursing, Hallym University, Chuncheon 24252, Korea; jayamush@hotmail.com

**Keywords:** health education, behavior change, prevention program

## Abstract

The present study aimed to systematically review to find the best available evidence on the efficacy of non-pharmaceutical interventions that have been used in the community so far. Through eight electronic journal database, 9 articles met our inclusion Participants, Intervention, Control, Outcomes, and Study Design (PICOS) criteria based on medical symptoms, interventions, and improvements. In general, interventions included hand hygiene, mask use, health education such as cough etiquette, hand washing and sanitizer methods. In addition, exercise and meditation were performed to improve immunity. As a result, the number of incidents and absences related to respiratory infections were reduced, the frequency and method of handwashing improved, and there were also positive effects in knowledge, attitude/perception, and performance. We concluded that it is necessary to create an environment and systematic support so that organizations or governments can determine healthy behavior at the same time as an individual approach. Furthermore, the follow-up for evaluating the effectiveness of interventions and the monitoring period should be included during the study, consequently resulting in having an opportunity to continuously remind people about health behavior. The community provides information on various types of non-pharmaceutical intervention to maintain healthy management and lifestyles in the public.

## 1. Introduction

Recently, many people have experienced a terrible global pandemic with coronavirus disease 2019 (COVID-19). COVID-19 has a similar presentation with respiratory infection in that it causes respiratory disease which presents as a wide range of illnesses from asymptomatic or mild through to severe disease and death [1]. However, the respiratory infection, including common cold, influenza, influenza-like-illness, especially seasonal influenza, is responsible for annual epidemics worldwide, consequently resulting in a significant public health burden [2]. It also continues to be a major cause of mortality in low-income countries [3]. In other words, because it is very common among the public and its speed of transmission and mortality is not as high as COVID-19 [4], substantive morbidity, mortality, and economic harms of the respiratory infection might be overlooked.

The World Health Organization (WHO) has recommended both pharmacological interventions and non-drug treatments for the prevention of infectious diseases [5]. A number of therapeutics have been developed so far and influenza vaccination is clearly the most important prevention strategy available [6]. Nevertheless, non-pharmaceutical interventions may also be important in the absence of sufficient vaccine supply and to reduce transmission of various respiratory viruses because the large number of immunotypes precludes the development of a vaccine [7]. Note that one major pharmacological intervention is often less effective among the elderly because of weakened immune systems, chronic disease complications, nutritional deficiencies, and lack of exercise, among other factors [8].

The respiratory infection is transmitted by direct with infected individuals, exposure to virus-contaminated fomites, and inhalation of infection aerosols [9]. As a result, some public health behaviors such as hand hygiene and appropriate respiratory etiquette have been considered as important actions to prevent the infection in terms of non-pharmaceutical interventions [10]. It can be explained that hand sanitizers and face-masks have been stockpiled during pandemic preparedness and are currently recommended in several countries. Data from a systematic review and a meta-analysis [11,12] showed that the hygiene behavior change by the handwashing with soap had been effective in reducing respiratory illness. Kawewchana et al. [8] evaluated effect of intensive hand washing education on the hand-washing behaviors in children, while using interactive participations including individual training, self-monitoring diary, and provision of soap. The authors confirmed improvement in both frequency and quality of hand washing. In addition, Nicholson et al. [13] provided a meaningful result in that direct-contact hand-washing interventions for the younger school-aged children can extend to affect the health of the whole family. Interestingly, even in military groups, hand-washing during field training was an effective precaution to significantly reduce incidence of the respiratory infection [7].

Measurement tools used in the non-pharmaceutical interventions in previous studies can be classified into medical, psychosocial, and health behaviors domains. For example, in the medical domain, incidence of respiratory infections, respiratory infection episodes, and inflammatory biomarker levels were used, whereas for the psychosocial domain, quality of life, social network, social capital, stress, and self-efficacy were utilized [5,7,9,10]. The frequency and quality of hand washing were adopted by Knowledge, Attitude and Practice (KAP) in the health behavior domain [2,4,6,8]

On the other hand, a study by Canini et al. [14] that evaluated the effectiveness of surgical facemasks to minimize influenza transmission by large droplets produced during coughing failed to confirm any significant effect. While being applied to as many as 509 participants, a study of Larson et al. [15] which compared three methods of education, education with alcohol-based hand sanitizer, and education with hand sanitizer and face masks also reported no significant difference among the methods. Even though they had large sample sizes, they concluded that there was no significant benefit of hand sanitizer and/or face masks over targeted education, except for reduced secondary transmission in the mask-wearing condition. Also, the results from Najnin et al. [16] supported the assertion that respiratory illness prevalence was similar between a control group and two experimental groups, i.e., cholera-vaccine-only, vaccine-plus-behavior-change such as handwashing promotion and drinking water chlorination.

Why does previous research show such a discrepancy in the effectiveness of non-pharmaceutical interventions for the respiratory infection? We strongly believe that intervention effect of the respiratory infection for public health might be changed according to target population, individual effort, method, and so on. Strikingly, Frieden [17] suggested a framework for public health action, while providing a five-tier pyramid consisting of socioeconomic factors at the base, changing the context to make individuals’ default decisions healthy as the second bottom level, long-lasting protective intervention in the third level, clinical intervention for the fourth, and counseling and education at the top. He argued that implementing interventions at each levels can achieve maximum possible sustained public health benefit [17]. However, based on the previous results, there are no definitive standards for what is the most effective non-pharmaceutical way to prevent respiratory infection and how it should be applied to the public and/or community. In terms of non-pharmaceutical and effective interventions, these studies have produced a problem, namely, how to organize and summarize their different findings, consequently resulting in the suggestion of needing a systematic analysis to achieve the highest level of evidence, which is also what we want to discuss in this review article. To guide and practice public health, the present study aimed to systematically organize what non-pharmaceutical interventions have been used in the community so far and what the results have been.

## 2. Materials and Methods

### 2.1. Strategy for Systematic Search

To provide transparency in systematic search and selection, the present study followed the methodology and protocols of the Preferred Reporting Items for the Systematic Reviews and Meta-Analysis (PRISMA) statement [18] and International Prospective Register of Systematic Reviews (PROSPERO; registered on doi:10.1186/2046-4053-1-5) part of the National Institute for Health Research (NIHR) [19]. Consequently, the study pre-established a data extract and analysis using inclusion and exclusion criteria for its systematic review procedure (Figure 1).

To search and identify articles for inclusion, 8 electronic journal databases, e.g., Embase, Medline, PubMed, Web of Science, Scopus, Cumulative Index to Nursing and Allied Health (CINAHL), SciELo, and Cochrane library were involved. The search period for published studies was set from January 2003 to the present, when related studies were actively conducted due to Severe Acute Respiratory Syndrome (SARS), which began in December 2002, using the key terms listed in Appendix A. The terms were divided into three groups: (1) medical symptoms; (2) interventions; and (3) improvements. The terms associated with each group were then listed by the Boolean term ‘OR’, and the resulting four groups were combined using the Boolean term ‘AND’. The final search was conducted on 10 September 2020.

### 2.2. Selection Criteria

All inclusion criteria were consistent in terms of Participants, Intervention, Control, Outcomes, and Study Design (PICOS) strategy [20]. The PICOS criteria are described in Table 1 below. Articles that did not correspond to topic, were not journal articles (letter, book, conference proceeding, only the abstract and pilot study), not peer-reviewed studies, or not written English, were excluded.

### 2.3. Selection Process for Studies

Based on combinations of the key terms (see Appendix A) in different databases, the articles with potential relevance were identified by carefully investigating the titles and abstracts, while removing duplicated articles. The two authors (S.P. and M.-S.K.) independently screened all searched articles for eligibility in terms of specific inclusion and exclusion criteria. In the case of any inconsistent review between the authors, they discussed the issue several times until the criteria were consistent and then selected the final analysis subjects. Subsequently, two authors (J.-S.Y. and M.-S.K.) assessed the full text of selected articles with a thorough compliance verification following the PICOS criteria. Then, the review and the systematic extraction of the data from the studies were included in the systematic review.

### 2.4. Assessment of the Quality of the Studies

To assess the quality of the selected studies, the NIH National Heart, Lung and Blood Institute (NHLBI) [21] guidelines were used in the present study. It was developed to assist reviewers in appraising and validating any potential flaws, including the sources of bias that existed in the study. Two authors (J.-S.Y. and M.-S.K.) independently evaluated the tools items as ‘yes’ (one point added) if the item criterion was clearly stated upon a literal reading or ‘no’ (no point added) if one or multiple criteria were missing or not applicable, could not be determined or not reported. After all the authors discussed the agreement on the evaluation, the final decision was made.

In detail, its scale criteria were (i) described as randomized, (ii) random allocation, (iii) treatment allocation, (iv) blinding (participant and provider), (v) blinding (assessor), (vi) similarity of groups at baseline, (vii) drop-out rate of 20% and less at end point, (viii) differential drop-out rate less than 15%, (ix) adherence to the intervention protocols, (x) similar background intervention, (xi) valid and reliable outcome measurement, (xii) power calculation, (xiii) pre-specified outcome (between groups), and (xiv) intention-to-treat analysis. Studies that scored 11–14 on the scale were considered to be ‘high’ quality methodologically. Scores ranging from 8 to 10 were ‘moderate’ quality, while scores ranging from 5 to 7 and below 4 were considered ‘low’ and ‘very low’ quality, respectively.

## 3. Results

### 3.1. Selection Outcomes

Using the search strategy, 40,017 records were identified in the 8 electronic journal databases. After removing 22,741 duplications, 17,276 articles were removed based on the exclusion criteria, such as not being peer-reviewed, not academic articles, and irrelevant topics. A total of 281 remaining articles that were potentially relevant articles were screened by carefully reading their titles and abstracts. Using these processes, 272 articles were excluded due to failed to PICOS criteria and then only 9 articles were included. Figure 1 visually displays each step.

### 3.2. Study Quality Scores

The scores for study quality, based on the checklists of NHLBI, were analyzed using a chi-square test and R statistical computing software. In Table 2, the study quality evaluation was provided for all nine studies. The authors confirmed that 3 of the selected studies had a ‘high’ quality. The remaining studies were evaluated as having a ‘moderate’ quality. The mean value of the study quality scores was 9.89 (standard deviation (SD): 1.05, range: 8~11). Then, to identify the goodness of fit for the study quality scores, a chi-square test was also conducted using the R software version 3.6.0 (R Foundation). There were no significant differences between the study quality scores (χ^2^ = 1.222, df = 3, *p* > 0.05).

### 3.3. Study Characteristics

#### 3.3.1. Participants and Study Setting

Table 3 summarizes the characteristics of the papers analyzed in this review in terms of participants and study setting, intervention, tools, and outcomes. According to our PICOS criteria, the participants were designated as the adults over 18 years of age. In detail, the age of the subject in the study of Barrett et al. [10] was specifically 30–69 years old, and the study conducted by Kim et al. [6] also reported subjects’ age as over 60 years old. However, there age of the subjects in the other studies was not accurately specified. Fortunately, since the place where the study was conducted was a military training camp [7], or a company worker [5,9], we assumed the subjects’ age. The study of Salmuna et al. [3] in which the targeted population was pilgrims was analyzed through the age suggested in the pilgrim qualification criteria. The subject criteria from the study of Kaewchana et al. [8] was households with influenza-positive children and their age was 6 years or older, but the average age displayed in the results was 30.

On the other hand, the places where the experiment was conducted were simply classified into the home, organization, community, and national levels. For the home, to evaluate the families with flu-infected children, the researchers visited households [8]. At the level of organization, the researchers met caregivers at the day care center for children [2]. They met soldiers admitted to training camps [7] and workers from various companies [5,9]. In the community level, the researchers recruited participants voluntarily as local residents [4,10], or older adults at various village halls in rural areas [7]. Interestingly, there was a study conducted in the Hajj Building Complex for people who went on a pilgrimage from one country to another [3].

The countries in which the study was conducted were: US [10], four in Europe (United Kingdom [4], Germany [9], Finland [5], Netherlands [2]), four in Asia (Malaysia [3], Thailand [8], two for the Republic of Korea [6,7]).

#### 3.3.2. Intervention

In general, many researchers provided a kit that included health education (hand hygiene, mask wear, cough etiquette, health practices for improving immunity), hand washing, exercise, meditation, and hand sanitizer at home, workplace, and community to prevent respiratory infections although the type of combination was different depending on the purpose and detailed characteristics of the study. Some stickers and/or posters were also used to remind people about the contents of the education. In addition, a message for a synchronization or self-regulation support program was sometimes provided to induce changes in individual behavior. Please see the detailed categories in Table 4.

While looking at the specifics of each intervention, we reorganized the intervention levels from the individual approach to the population approach. First, in the individual approach, the personalized synchronization message and self-regulation support program were organized into four sections based on web. The first section provided all the essential components of the intervention, including information on the medical team. For examples, it noticed any relation between handwashing and viral transmission, expert recommendations on handwashing frequency and techniques, how to pick up free handgels at local clinics, while providing personalized feedback for participants to develop and improve handwashing plans. Then the users printed and signed. Also, they made the plan and encouraged the involvement of other family members. The remaining three sections were designed to reinforce positive attitudes and norms [4].

Intensive handwashing education at the home level consisted of a 30-min discussion, personal training, self-monitoring diary, and preparation of soap and materials. It was repeated on the 3rd and 7th training days. The control group was educated for 30 min on general influenza infection, nutrition, physical activity and smoking cessation [8].

Based on the institution level, the company’s workers were educated by specific methods while being given a Handrub 500 cc product made of alcohol and hand care lotions; “wet your hands sufficiently with a hand sanitizer to rub your hands and use them five times a day”. Also, they were educated to do hand friction; “After using the toilet, when touching your nose, before eating, and after touching sick coworkers, customers, and some materials, you should do hand rubbing” [9]. Another study of workers compared between a group who used soap and water along with personal hygiene education and a group who used hand sanitizer containing alcohol [5]. The day care center caring for children gave hand hygiene products to caregivers for 6 months, trained in the hand hygiene guidelines of the Netherlands, and conducted education to set goals and formulated specific hand hygiene improvement activities. In addition, posters and stickers were used to specifically write the hand hygiene improvement activities to recall users’ actions and clues [2]. Kim performed handwashing by supplying water and a basin during field training for the military’s 4-week basic trainers, and promoted the ‘good handwashing method’ on the bulletin board at training centers and toilets [7].

The educational intervention for the prevention of respiratory infections in the studies of the community level was conducted for the elderly aged 60 or older at the town hall, and once per a week (total five times), 50 min at a time. Specific contents included cough etiquette, hand washing, oral health, and exercise [6]. In addition, exercise and meditation for local community residents were conducted for 14 to 16 people per class for 8 weeks each, 2.5 h per class, and each person was allowed to practice for 20–40 min each day at home, and an additional 5 h for each program. The weekend retreat was held [10]. Finally, at the national level, a hand hygiene kit (4 bottles of hand rubbing agent) was provided to the pilgrims, and education and demonstration on the correct method and eating habits of hand rubbing and mask wearing on a one-to-one basis, and a pamphlet on hand rubbing methods were provided [3].

#### 3.3.3. Tools

In this review study, the measurement tools used in relation to respiratory infection prevention included disease incidence, absenteeism or sick leave due to disease, frequency of hand washing, quality assessment, compliance and intention, and KAP for hand washing. External psychosocial domains and physical activity assessments were used. For example, there were an incidence of diseases or symptoms related to respiratory infections [5,7,9,10], absenteeism due to disease and sick leave rate [5,9], and a virus test and blood test which were performed directly through an inflammatory biomarker related to ARI (acute respiratory infection) [10]. In the study conducted by Barrett et al. [10], a moderate assessment of ARI disease was measured by daily self-report using the Wisconsin Upper Respiratory Symptom Survey (WURSS-24), and the number of ARI incidences and disease severity were evaluated by the pre-intervention group. It was used to identify characteristics of each study.

Usually, the kind of measurements used for hand washing included its frequency [4,8], its technique (quality evaluation) [8], its compliance [2,9], and intention to wash hands [4]. Certain standards for handwashing were suggested and implemented in compliance with regulations. In particular, quality evaluation of the handwashing [8] was evaluated by handwashing technology, while using four types of soaping practice, rubbing the hand area, duration, and drying method as a total of 8.5 points. The points were calculated as 1 point for use of soap and 5.5 points for when all 7 parts of the handwashing (e.g., palm, back of hand, fingers, crossed fingers, fingertips, thumb, and wrist) were implemented. Also, one point was given for rubbing hands for more than 20 s, and 1 point for drying the hands with a clean towel or paper towel.

Measurements of KAP evaluated knowledge, attitudes, and practice related to respiratory infections and influenza prevention [3,6,8]. In the study of Kaewchane et al. [8], knowledge (5 questions), attitude (5 questions), and practice (5 questions) tools for influenza developed by the researcher were used. Kim et al. [6] used a knowledge (10 questions), attitude (10 questions), and performance (10 questions) questionnaire for respiratory infection prevention developed by Kwon and Yu [22]. Salmuna et al. [3] had 10 questions on three domains risk factors and ILI (influenza-like illness) transmission method for knowledge, a prevention method of ILI for perception, and respiratory infection prevention consisting of dietary habits, physical activity, and hand hygiene for ILI prevention, while also using the Respiratory Infection Preventive Measure Questionnaire, KPP-PMQ) questionnaire.

First, in the measurement of the psychosocial domain, Barrett et al.’s study [10] used general mental and physical health (SF-12), perceived stress (PSS-10), sleep quality (PSQI), self-efficacy (MSES, ESES), mindful awareness (MAAS), positive and negative emotion (PANAS), perceived social support (SPS), and the sense of feeling loved, five important personality traits (BFI), and the social network (SNI), and the Kim et al. study [6] used social capital. In addition, Yardley et al.’s study [4] included intention, attitudes, subjective norms, and perceived behavioral controls and risk of infection based on the planned behavioral cognitive theory. For measurement of physical activity, the Global Physical Activity Questionnaire (GPAQ) was used in the study of Barrett et al. [10].

#### 3.3.4. Outcomes

In this study, the effectiveness of the intervention program on the prevention of respiratory infections was determined by the incidence of disease, absenteeism or sick leave due to disease, frequency of hand washing, quality evaluation, compliance and intention, and also included the results of psychosocial and physical activity evaluation.

For the disease incidence rate, the cumulative rate of total ARI incidence decreased during the 4-week period in the early handwashing group and also at 2 weeks after the intervention in the study of Kim et al. [7]. In the study of Hübner et al. [9], the incidence of common cold, fever, and coughing were decreased. In the study of Savolainen-kopra et al. [5], the incidence of ARI infection decreased by 6.7% over the entire study period. The incidence of absenteeism or sick leave showed a tendency to decrease similar to disease incidence in Hübner et al. [9], but diarrhea showed a statistically significant difference, and bronchitis showed a significant difference in the control group. On the other hand, in Savolainen-korpa et al. [5], no reduction in the incidence of absenteeism or sick leave was observed. In a study by Barrett et al. [10], the biomarker showed an increase during ARI, and the time-wise comparison within each group showed a significant decrease in CRP in the meditation group and IP-10 in the exercise group.

The average frequency of handwashing was found to be higher in the intervention group [4,8], and the handwashing quality evaluation [8] improved more after 90 days than on the 7th day after the intervention. Although there was no statistically significant difference in handwashing compliance in Hübner et al. [9], the rate of practice at least three times a day was 78.8%. In the study of Zomer et al. [2], compliance was also increased in the intervention group and decreased during the 6-month observation period in the control group. In addition, it was found that the intention to wash hands increased more in the intervention group [4].

In the KAP assessment related to respiratory infection prevention, knowledge, attitude, and practice were all improved in the study of Kim et al. [6]. Also, knowledge was improved and some items related to hand washing in the attitude and performance were significant differences in the study of Kaewchana et al. [8]. However, in the study of Salmuna et al. [3], perception scores were significantly decreased in the post-test compared to the scores of pre-test although the scores of the knowledge and performance between two tests were not differ.

Other psychosocial areas were statistically significantly improved in general mental health, self-efficacy, attention, sleep quality, perceived stress and depressive symptoms in the MBSR (mindfulness-based stress reduction) and EX groups compared to the control group in the Barrett et al.’s study [10]. In a study by Kim et al. [6], social capital was found to be improved. In addition, it was found that the intervention group significantly improved in terms of attitude in the planned behavioral cognitive theory, and showed a significant positive indirect effect on the change in handwashing through intention and attitude as a factor affecting cognitive behavior. In the evaluation of physical activity, there were significant differences according to the measurement period (4 measurements) in the exercise intervention group [10].

## 4. Discussion

The health and socioeconomic impact of influenza worldwide is substantial and underscores the importance of improving influenza control measures. The purpose of the present study was to systematically organize what non-pharmaceutical interventions have been used in the community so far by the systematic review and what the results have been in terms of appropriate intervention methods, effective intervention duration, and reliable tools and data.

### 4.1. Have Various Interventions Been Conducted to Prevent Respiratory Infection in the Previous Studies?

While looking at the intervention methods provided in the studies as a personal approach., there were several health educations: informative program related to respiratory infection prevention (handwashing, cough etiquette, etc.) and personal hygiene; training, exercise, and meditation to enhance immunity (mindfulness meditation); informative program [6,10] using the web-based education to reduce from infection risk (such as healthy lifestyle habits, health food consumption, etc.) and to strengthen the immune system [4]. These health education interventions represent the highest level in the five stages of the health impact pyramid for public health behavior change, and correspond to an individual approach, and the effectiveness of interventions tends to be somewhat lower than those of levels 1–4 targeting a large population. As the most commonly used method, this is a useful intervention that, if applied consistently and repeatedly, can affect changes in individual health behavior [17]. Moreover, as individual lifestyles are regulated through interactions with social environments [23], changes in individual health behaviors through education can affect the individual’s immunity [17].

On the other hand, for homes and workplaces including the military group, personal hygiene products such as soap and alcohol gel for hand sanitization were provided in the toilet and wash basin during the study. When attaching a poster or sticker on the handwashing method, people were educated as an environmental approach. In particular, use in public places of health-related promotional materials such as posters, stickers, and pamphlets can be used as an effective approach to promoting personal hygiene practices through repeated exposure [24]. It can be seen in the same context as the second stage of the pyramid, as far as it affects the health behavior of large population groups by changing the environmental situation through social/political intervention [17].

Also, it means that organizations such as family and community that share and interact with each other, or densely populated work places, schools, and the military are not only able to intervene at the individual level, but also to the environment [22]. We believe that these changes (e.g., provision of personal hygiene products, composition of diets to increase immunity, establishment of systems to improve physical activity, establishment of health villages, etc.) can have a significant impact on health improvement at the local or organizational level. In particular, such interventions are at the organizational or governmental level with individual approaches, such as providing health education and hand hygiene kits to religious groups that require movement between countries, i.e., pilgrims participating in large-scale religious events [3]. It will be necessary to create an environment systematically and to provide specific guidelines so that healthy behavior can be determined in the community.

### 4.2. What Is the Appropriate Duration of Intervention for Health Behavior Change?

According to the trans-theoretical model, the stage at which health behavior changes begin and the period of maintenance is 6 months [25]. In other words, at least 6 months of intervention is required to improve and maintain the health level of individuals or communities and to verify the effectiveness of the program. The duration and frequency of interventions analyzed in the present study ranged from at least 1 to 8. For example, One-time single intervention was the most common in 5 studies out of 9 papers [3,5,7,8,9]. One of them provided similar content repeatedly up to three times through home visits [8]. Intervention was guided by suggesting criteria for handwashing and personal hygiene use. In addition, an integrated program was offered according to a single topic related to respiratory infection prevention: one 3-session program [2] and two 4-session programs [4,6]. An integrated program including cough etiquette, oral hygiene-including brushing teeth, mask usage, hand washing, walking exercise, healthy lifestyle habits, health food consumption, etc.) was provided. Exercise and meditation programs for improving immunity were also practiced for 8 weeks [10].

Among the sessions provided up to 15 sessions in a review of intervention programs for health behavior control, the 6–10 session program showed the strongest effect [26]. On the other hand, the study of Kim and Yang [27] reported that the program effect was greater with a maximum of 21 sessions, and the greater the frequency of provision, but the papers reviewed in this study provided a somewhat lower frequency than the previous studies with only a maximum of 8 provisions. Nevertheless, when the results of these studies are summarized, it was confirmed that the number of infectious diseases was lowered and handwashing compliance (practice rate) was also improved. Thus, it was necessary to set an appropriate period based on the cost-effectiveness [28].

While looking at the evaluation types used in all studies, 2 studies performed only the pre-posttest without follow-up monitoring [3,7], and the remaining 7 studies performed the evaluation at 3 months after intervention [4,8], 6 months [2,6,10], 12 months or longer [5,9]. In the form of intervention for follow-up, one additional reinforcement training summarizing the existing program was provided [6], and hand hygiene products such as soap and hand sanitizing gel were provided [5,8,9]. Also, for self-reporting, regular direct contact was made via telephone or e-mail [5,9]. As a result, handwashing compliance (practice rate) was improved in the final evaluation compared to the baseline [2,4,8,9], and the number of infectious diseases was decreased [5,7]. The KAP score for respiratory infection prevention was also improved [6]. However, a study that provided a hand hygiene kit after one training session and conducted an evaluation of knowledge, perception and practice without additional intervention showed a significant difference in perception, but it was not effective because it was found that the recognition score decreased further after the intervention [3].

In summary, although formal interventions are not very frequent, follow-up for evaluation or interventions through monitoring lasted from 1 month to 15–16 months. During this period, roles that remind people of the actions to be performed in any activity will have a continuing influence on changes in individual behavior. Therefore, in order to determine the duration of intervention, it is necessary to include both the post-intervention monitoring and follow-up periods while considering the limited cost, time, and space.

### 4.3. Have Data Been Objectively Collected by Using Reliable Methods?

The measurements reviewed in this paper can be classified into three types: self-report [4,8,9] and observational methods [2,8], physiological measurement [5,10].

In detail, self-report has the advantage of reflecting individual attributes well and being able to quickly collect data, but may threaten the reliability and validity of the measurement due to the response bias of participants [29]. Studies were measured by self-report for knowledge, attitude (or perception) and practice [3,6,8], psychosocial domain [10], and physical activity evaluation [10], and respiratory infection-related disease incidence rate [7,9]. Among them, the KAP tool was developed with an item that fits the characteristics of the subject and presented the evidence by verifying the content validity and reliability in two studies [6,8]. However, the suggested reliability was lower than the acceptable reliability coefficient of 0.70~0.90 [30] by more than 0.50 and less than 0.70.

In addition, when using an existing tool in the form of a checklist [10] and when similar symptoms for respiratory infections occur, the measurement of the incidence of respiratory infection-related diseases was performed by medical staff [7]. Self-reported measurement is a form of data collection with prominent subjective characteristics of participants, so it is necessary to use a validated tool for reliable research results suitable for the purpose of the research. It is also very important to secure objectivity for the collected data.

Second, the observation method was useful for measuring behavioral change [29], and had less effect on the observer. However, since the observer’s bias can be reflected [31], measures to maintain the reliability and objectivity of the observer must be considered together [29]. The observation method was mainly used to measure handwashing frequency (practice rate) and compliance (according to guidelines) [2,8]. Participants were educated on handwashing method guidelines and were to follow the guidelines, and observers were educated on measurement guidelines and then evaluated accordingly. However, there was only one study that provided the source for the handwashing guidelines, and no evidence was found for verifying the reliability between observers among the studies reviewed [2]. As mentioned earlier, since observation methods tend to be subjective compared to other types of measurement methods, reliability may be inferior [32], so securing reliability between observers is very important.

Lastly, there is a measurement method using biomarkers as physiological indicators. The physiological indicator measures the physical and biological state of the subject, and the objectivity of the collected data is high, and it is a measurement method with high reliability compared to other social psychological measurement methods. However, expensive specific measurement equipment must be provided, and an accurate understanding of the use of the equipment is required [32]. In two studies [5,10], a biomarker test for infection symptoms was conducted, but specific information on the analysis method or analysis equipment was insufficient as the basis for the accuracy or reliability of the measurement. Nevertheless, by measuring physiological indicators according to the self-report results of respiratory symptoms, it will provide evidence to increase objectivity for subjective data.

In summary, the effectiveness of the intervention was confirmed through the measurement tools used in each study, but studies using the validated tools to secure the reliability or validity of the results were limited to some. Moreover, only three studies have used two or more of the methods for data collection [5,8,10]. In the case of the self-reporting method based on the KAP model reviewed in this study, behavior change can be according to the theoretical basis that a high level of knowledge affects attitudes and performance [6,33]. Therefore, if we measure the degree of actual performance, including observation, it will be a way to increase the objectivity of the study. As such, it is necessary to consider a multi-method approach including objective performance measurement [34], on the possibility of underestimating or overestimating the effectiveness of interventions depending on the data collection method. Therefore, in order to secure reliable and high-powered results, the use of evidence-based verified tools and an approach using various data collection methods are required. 

### 4.4. Limitation of the Study

In this systematic review, meta-analysis which is regarded as a highly evaluated evidence-based study was not performed. Data reported in individual studies could not be extracted due to the different units of measurement, and the number of randomized controlled trial studies that met the PICOS criteria were insufficient to perform the meta-analysis. Therefore, the effect size of data from each article could not be compared with the meta-analysis method. In other words, the most effective non-pharmaceutical intervention of respiratory infection among various interventions could not be analyzed statistically in this study due to lack of data that can be extracted. As the same view, previous researchers who conducted the studies had so many different methodologies, and then their results measured from dissimilar perspectives had limitations in drawing clear research questions and appropriate conclusions in the present study.

Another limitation is that the measurements reported in many studies were self-report outcomes. Fewer studies have reported results of objective testing tools. Self-reporting that is not provided complementarily with the result of objective tools could overestimate the post-test results due to pretest sensitization [35]. It is essential to develop a tool that can measure the effect of intervention objectively for better and accurate comparison.

## 5. Conclusions

This study conducted a systematic literature analysis to find specific strategies for what is the most effective non-pharmaceutical intervention to prevent the respiratory infections and how it should be applied to the public and/or community. Through the review, it was found that it is necessary to create an environment and systematic support so that organizations or governments can determine healthy behavior at the same time as an individual approach. In addition, as a specific intervention strategy, even if the frequency of interventions is low or the actual duration of interventions is short, the follow-up for evaluating the effectiveness of interventions or the monitoring period should be included during the study, consequently resulting in having an opportunity to continuously remind people about health behavior. Data collection should use various methods of data collection that can derive objective and reliable outcomes, such as the self-reporting format, observation method, and biomarkers, and use measurement tools with high reliability and validity. In conclusion, it is necessary to perform various types of non-pharmaceutical intervention to maintain personal hygiene management and healthy lifestyles in the community.

## Figures and Tables

**Figure 1 ijerph-18-03927-f001:**
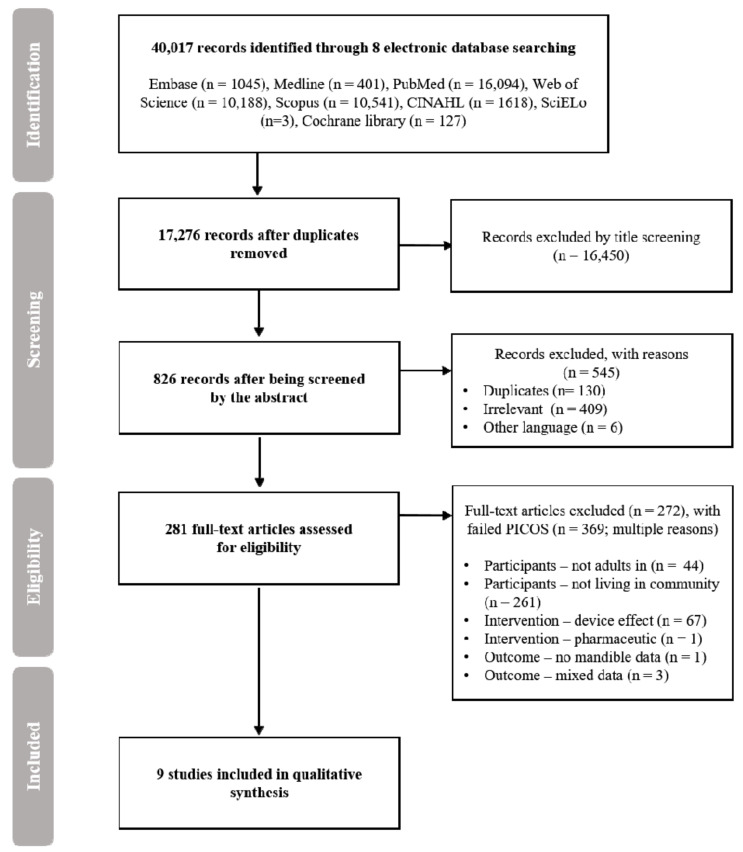
A Preferred Reporting Items for the Systematic Reviews and Meta-Analysis (PRISMA) chart to describes study selection process using the inclusion and exclusion criteria for the present study. PICOS: participants, intervention, control, outcomes, study design.

**Table 1 ijerph-18-03927-t001:** Inclusion criteria based on Participants, Intervention, Control, Outcomes, and Study Designs (PICOS) strategy in the present study.

Parameter	Inclusion Criteria
Participants	Adults living in the community (aged 18+)
Intervention	Non-pharmaceutical interventions (hand washing, personal hygiene, exercise, cough etiquette, nutrition, oral health, sleep, meditation, etc.)
Control	Comparison to a control group or repeated measures (pre- and post- intervention comparison)
Outcomes	Respiratory infectious disease incidence rate (acute respiratory infectious disease incidence rate, influenza outbreak, ILI (influenza-like disease): sneezing, runny nose, sore throat, fever, muscle pain, etc.) Hospitalization rate due to respiratory disease Changes in health behavior (mask wearing, proper hand washing, personal hygiene, sleep, exercise, etc.) KAP (knowledge, attitude, practice) measure Social capital Self-efficacy/Confidence Health behavior (mask wearing, proper hand washing, personal hygiene, sleep, exercise, etc.)
Study Designs	Integrated study design of randomized controlled trials, non-randomized controlled trials, cohort studies, and repeated measures (experiments with additional purposes) to report the results of pre- and post- intervention.

**Table 2 ijerph-18-03927-t002:** Quality assessment of randomized control-trial based on the National Heart, Lung and Blood Institute guidelines for enrolled studies [21].

Study	1	2	3	4	5	6	7	8	9	10	11	12	13	14	Total	Quality
Barrett et al. [10]	1	1	1	0	0	1	1	1	1	1	1	0	1	1	11/14	High
Hübner et al. [9]	0	0	1	0	0	1	1	1	1	1	1	0	1	1	9/14	Moderate
Kaewchana et al. [8]	1	1	1	0	0	1	0	1	1	1	1	1	1	1	11/14	High
Kim et al. [6]	0	0	1	0	0	1	1	1	1	1	1	1	1	1	10/14	Moderate
Kim et al. [7]	0	0	1	0	0	1	1	1	1	1	1	0	1	1	9/14	Moderate
Salmuna et al. [3]	1	1	1	0	0	0	0	1	1	1	1	0	0	1	8/14	Moderate
Savolainen-Kopra et al. [5]	1	1	1	0	0	1	1	1	1	1	0	0	1	1	10/14	Moderate
Yardley et al. [4]	1	1	0	0	1	1	1	1	1	1	1	0	1	1	11/14	High
Zomer et al. [2]	1	1	1	0	0	0	1	1	1	1	1	1	1	0	10/14	Moderate

Scale of item scores: 0 = absent; 1 = present. The NIH scale criteria were (1) Described as randomized; (2) Random allocation; (3) Treatment allocation; (4) Blinding (both participant and provider); (5) Blinding (assessor); (6) Similarity of groups at baseline; (7) Drop-out rate 20% and less at endpoint; (8) Differential drop-out rate less than 15%; (9) Adherence to intervention protocols; (10) Similar background intervention (11) Valid and reliable outcome measurement; (12) Power calculation (13) Pre-specified outcome (between groups); (14) Intention-to-treat analysis. Studies’ scoring 11–14 on the scale were considered to be “high” quality methodologically. Scores’ ranging from 8 to 10 were “moderate” quality, and studies’ scoring 5 to 7 were “low” quality; studies’ scoring below 4 were considered “very low” quality.

**Table 3 ijerph-18-03927-t003:** Characteristics and training outcomes for all enrolled 9 studies based on PICOS criteria.

Study, Country	Design	Setting	Participants		Intervention		Tools	Outcomes
			Experimental	Control	Type	Session (Period)		
Barrett et al., USA [10]	RCT	Community	EG1: *n* = 137 (mean 49.1 years) EG2: *n* = 138 (mean 49.2 years)	*n* = 138 (mean 50.7 years)	EG1: Progressive moderate intensity exercise program EG2: Mindfulness-based stress reduction program	8 sessions (2.5 h per week, home practice 20–45 min per day) and 1 session (retreat, 5 h)	ARI illness episodes Psychosocial domains: SF-12, PSS-10, PSQI, MSES, ESES, MAAS, PANAS, SPS, BFI, SNI, Seattle Index, The sense of feeling love GPAQ Inflammatory biomarker levels Follow up: Baseline, after intervention 1–2, 3, 4–5, 6 months	Decreased ARI illness episodes Improved MAAS, PSS10, PSQI, GPAQ, MSES, ESES and PHQ9 in EG1 Improved MAAS, SF12, PSS10, MSES and ESES in EG2 Decreased CRP in EG2, and IP-10 in EG1 on biomarkers
Hübner et al., Germany [9]	RCT	Work place	*n* = 64 (mean 43.6 years)	*n* = 65 (mean 45.6 years)	Educated hand hygiene method Provided a handrub gel	1 session (time: no report)	Illness and absence episodes Follow up: Monthly self-report until 12 months	Reduced illness episodes (related to common cold, fever, coughing) Reduced absent episodes (related to diarrhoea) Reduced total number of day ill(related to colds, fever, cough)
Kaewchana et al., Thailand [8]	RCT	Home	FA: *n* = 240 (mean 33.8 years) QA: *n* = 164 (mean 35.7 years)	FA: *n* = 135 (mean 45.6 years) QA: *n* = 166 (mean 34.8 years)	∙Educated hand washing (home visiting on 3 and 7 days) Provided soap & Hand wash poster	1 session (30 min, provide repeat up to 3 times)	Hand washing frequency Hand washing quality Knowledge, Attitude, Practice Follow up: Baseline, after intervention 7 days, 90 days	Increased hand washing frequency Improved hand washing quality Improved Knowledge, but only improved partial items in the Attitude, Practice
Kim. et. al., South Korea [6]	Quasi- experimental	Rural community	*n* = 37 (mean 76.6 years)	*n* = 32(mean 74.4 years)	Respiratory infection preventive education program based on social cognitive theory	4 sessions (50 min per sessions, session per week) and 1 session (for reinforcement)	Knowledge, Attitude, Practice Social capital Follow up: Baseline, after intervention, 3 months, 6 months	Increased Knowledge, Attitude, Practice and Social capital
Kim et al., South Korea [7]	Quasi- experimental	Military training facility	*n* = 631 (age: no report)	*n* = 660(age: no report)	Educated hand washing method Attached a hand washing poster	1 session (time: no report)	ARI episodes Follow up: pre-post test	Reduced ARI episodes
Salmuna et al., Malaysia [3]	RCT	Hajj building complex	*n* = 94 (mean 51.2 years)	*n* = 78	Health education Provided handrup gel (4 bottles) and pamphlets	1 session (time: no report)	Knowledge, perception, practice Hand washing compliance (comply to handrup usage or not) Follow up: pre-post test	Decreased Perception and no change in the Knowledge, Practice
Savolainen- Kopra et al., Finland [5]	RCT	Work place	EG1: *n* = 257 (mean 45.1 years) EG2: *n* = 202 (mean 42.7 years)	*n* = 224 (mean 42.8 years)	Guidance how to respiratory infection prevention Provided a liquid hand soap (all groups) or alcohol-based handrup (EG2) Handwashing type by the trial group -EG1: Soap and water wash -EG2: Alcohol-based handrup	1 session (time: no report)	Respiratory infection, sick leave, absence episodes Follow up: Weekly self-report until 15–16 months	Reduced infection episodes in EG1
Yardley et al., United Kingdom [4]	RCT	Community	*n* = 336 (mean 49.2 years)	*n* = 336 (mean 50.9 years)	Web-based education of tailored motivational message and self-regulation support	4 sessions (session per week, time: no report)	Hand washing frequency Planned behavior cognition: Intentions, Attitude, Subjective norm, Perceived behavior control Follow up: Baseline, after intervention (4 weeks), 12 weeks	Increased hand washing frequency Improved hand wash intentions and attitude Revealed positive indirect effects on changes in handwashing via intentions and attitude
Zomer et al., Netherlands [2]	RCT	Day care centers for children	*n* = 36 (DCC) * total caregivers: *n* = 795	*n* = 35 (DCC)	Educated Dutch national hand hygiene guidelines Provided a hand hygiene products (paper towers, soap, alcohol- based hand sanitizer and hand cream) and information booklet (outlining train contents)	3 sessions (period and time: no report)	∙Hand hygiene compliance (according to guidelines) ∙Follow up: Baseline, after intervention 1 month, 3 months, 6 months	∙Increased hand hygiene compliance

PICOS: participants, intervention, control, outcomes, study design; RCT: randomized controlled trials; ARI: acute respiratory infection; EG: experimental group; SF12: medical outcomes study short form; PSS-10: perceived stress scale; PSQI: Pittsburg sleep quality index; MSES: mindfulness self-efficacy scale; ESES: exercise self-efficacy scale; MAAS: mindfulness attention awareness scale; PANAS: positive and negative affect schedule; SPS: social provisions scale; BFI: big five inventory; SNI: social network index; GPAQ: global physical activity questionnaire; PHQ9: patient health questionnaire-9; CRP: C-reactive protein; IP: interferon gamma-induced protein; MBSR: mindfulness-based stress reduction; FA: frequency assessment; QA: quality assessment; RIPEP-SCT: respiratory infection preventive education program based on social cognitive theory; DCC: daycare centres. * Total cumulative caregivers who participated until the 3rd follow-up period.

**Table 4 ijerph-18-03927-t004:** Types of intervention in enrolled studies.

Studies			Barrett et al. [10]	Hübner et al. [9]	Kaewchana et al. [8]	Kim et al. [6]	Kim et al. [7]	Salmuna et al. [3]	Savolainen-Kopra et al. [5]	Yardley et al. [4]	Zomer et al. [2]
Types of Intervention											
Health education	Hand hygiene		-	v	v	v	v	v	v	v	v
	Coughing etiquette		-	-	-	v	v	-	v	-	-
	Wearing a mask		-	-	-	v	-	v	-	-	-
	Oral hygiene		-	-	-	v	-	-	-	-	-
	Improving immunity	Exercise	v	-	-	v	-	v	-	-	-
		Meditation	v	-	-	-	-	-	-	-	-
		Nutrition	v	-	-	-	-	v	-	-	-
		Etc.	-	-	-	-	-	v ^a^	-	v ^b^	-
Hygiene products	Soap		-	-	v	-	v	-	v	-	v
	Handrup gel		-	v	-	-	-	v	v	v	v
	Hand cream		-	v	-	-	-	-	-	-	v
	Etc.		-	-	-	-	-	-	-	-	v ^c^
Promotional materials	Posters/Stickers		-	-	v	-	v	-	-	-	v
	Pamphlets/Booklet		-	-	v	-	-	v	-	-	v

^a^ Smoking cessation, ^b^ Taking Echinacea, ^c^ Paper towels.

## Data Availability

Data sharing is not applicable to this article.

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
