# Peer review of "A Systematic Review for Effective Preventive Public Education of Respiratory Infection"

_ijerph, 2021, doi:10.3390/ijerph18083927_

Round 1

Reviewer 1 Report

The article is useful in the field of Civic Education of the population in periods of pandemic or public health outbreaks and leaves open important elements on social behaviour and healthy lifestyles in the community.
This article is framed in the concept of Social Health defined by WHO and with a good methodological proposal and technical tools for a systematic intervention in public health.
I only have reservations about the title of the article "non-pharmaceutical intervention", because it does not have a common understanding at the international level. I consider that the article is very focused on Social Health with a strong focus on preventive social education of the population with regard to individual, collective and community social behaviour.
The non-pharmacist reduces the analysis to non-drug habits and in my reading the article does not focus on this focus.
The article, although deductive, makes a first contribution to the analysis of preventive epidemiological group and community practices and this only reinforces the social dimension of public health.

Author Response

Dear, Reviewer

Thank you for your opinion.

Please see the attachment for the details about response.

sincerely,

Myung Soon Kwon

Reviewer 2 Report

Proposed paper is interesting and well written. I find it useful to analyze the non-pharmacological interventions used in the community. But I have several comments:

Point 1: In the introduction the importance of respiratory infections, the mode of transmission… are introduced and explained however, how it is measured i.e. the tools used to measure it are not described at all. I think it would be useful to provide some explanation.

Point 2:  In materials and methods (line 105) the registration code in PROSPERO should be added.

Point 3: Could the authors explain why they ended with only 9 publications when 17,276 publications were selected with the chosen keywords? May be the keywords are not adequate as it is written that 16,450 were irrelevant records. Therefore, may be other interesting data have been published but not analyzed.

Point 4: Line 171 “The authors confirmed to agree that…” In the results section, only the results obtained should be presented, the rest in the discussion.

Point 5: In section 3.3.1 Study Characteristics, it would be useful to refer to Table 3.

Point 6: I find the first paragraph of section 3.3.2 confusing, I think the wording should be changed. It seems that in all the researches the same intervention was made.

Point 7: Line 297-303 I also find this paragraph confusing as to which studies use which instruments. Perhaps it could include in line 302 “BFI AND the social network in the study of Barret…” to make it clear that the above measures are those used in the Barret study and social capital in the Kim study. Or change the wording.

Author Response

(The authors gave the same response as above.)

Reviewer 3 Report

A Systematic Review for Effective Non-Pharmaceutical Inter-2 vention of Respiratory Infection in the Community

Abstract

Regarding:  The study aimed to systematically organize what non-pharmaceutical interventions have been used in the community so far and what the results have been in terms of appropriate methods, effective duration, and reliable tools for the data collection.

This tends to confuse (introduces a lot of noise and does not allow for precision): of appropriate methods, effective duration, and reliable tools for the data collection.

SUGGEST REDACT: Realize a systematic review to find the best available evidence on the efficacy of non-pharmaceutical interventions that have been used in the community so far.

PICO is a methodology: Through the PICO methodology the guiding question of the study is constructed............

The question should account for the intervention compared to no intervention or compared to another pharmacological intervention.  The important thing is to analyze the effect of the intervention or several interventions for which several separate searches (for each intervention) would be required.

Methodology: The researchers used the PRISMA (Preferred Reporting Items for the Systematic Reviews and Meta-analysis) method and the PROSPERO (International Prospective Register of Systematic Reviews) of the National Institute for Health Research (NIHR) recommended for the approach of systematic reviews.

Researchers justify the search period (2003 to date). This is because related studies were actively conducted due to Severe Acute Respiratory Syndrome (SARS), which began in December 2002.

However, the number of articles found indicates that the topic has not been adequately studied or the precision of the search does not allow for other publications that could contribute to answering the guiding question of the study.

Regarding the guiding question of the study, it is not clear. There is confusion among the various interventions that are obviously approached from different methodologies that do not allow to be analyzed by means of the criteria established by the authors. The result of the search and the limitations of the study is another way of demonstrating that there is no precision due to the lack of an adequate research question and a defined outcome.

However, the topic is interesting and perhaps researchers could be encouraged to reorient the results and ways of presenting the summary table. Table 3 is very confusing as there is too much information and each one oriented to different outcome.

SUGGESTION. Construct a table of results for each intervention. For example: Hygiene Education

It is likely that some of the discards could be useful and provide information about them. Of course, with the proper care of the quality of the methodology and biases.

Author Response

(The authors gave the same response as above.)

Round 2

Reviewer 2 Report

I think the paper is clearer. Congratulations!

Just add that on line 237 the doi refers me to another article. Could you provide the registration number?

Reviewer 3 Report

I have read the second version of the manuscript A Systematic Review for Effective Preventive Public Education of Respiratory Infection. I consider that the authors have taken into account the suggestions that were indicated. The tables consider information that clarifies their content and interpretation for the readers of the journal. I have no further comments